# Development of Molecular Markers for Detection of *Acidovorax citrulli* Strains Causing Bacterial Fruit Blotch Disease in Melon

**DOI:** 10.3390/ijms20112715

**Published:** 2019-06-02

**Authors:** Md. Rafiqul Islam, Mohammad Rashed Hossain, Hoy-Taek Kim, Denison Michael Immanuel Jesse, Md. Abuyusuf, Hee-Jeong Jung, Jong-In Park, Ill-Sup Nou

**Affiliations:** Department of Horticulture, Sunchon National University, Suncheon, Jeonnam 57922, Korea; rafiq@pstu.ac.bd (M.R.I.); m.r.hossain@bau.edu.bd (M.R.H.); michaelijesse@gmail.com (D.M.I.J.); yusuf_agr@pstu.ac.bd (M.A.); gml79wjd@sunchon.ac.kr (H.-J.J.); jipark@sunchon.ac.kr (J.-I.P.)

**Keywords:** *Acidovorax citrulli*, bacterial fruit blotch, melon, molecular marker, polymerase chain reaction, whole-genome alignment

## Abstract

*Acidovorax citrulli* (*A. citrulli*) strains cause bacterial fruit blotch (BFB) in cucurbit crops and affect melon significantly. Numerous strains of the bacterium have been isolated from melon hosts globally. Strains that are aggressively virulent towards melon and diagnostic markers for detecting such strains are yet to be identified. Using a cross-inoculation assay, we demonstrated that two Korean strains of *A. citrulli*, NIHHS15-280 and KACC18782, are highly virulent towards melon but avirulent/mildly virulent to the other cucurbit crops. The whole genomes of three *A. citrulli* strains isolated from melon and three from watermelon were aligned, allowing the design of three primer sets (*AcM13*, *AcM380*, and *AcM797*) that are specific to melon host strains, from three pathogenesis-related genes. These primers successfully detected the target strain NIHHS15-280 in polymerase chain reaction (PCR) assays from a very low concentration of bacterial gDNA. They were also effective in detecting the target strains from artificially infected leaf, fruit, and seed washing suspensions, without requiring the extraction of bacterial DNA. This is the first report of PCR-based markers that offer reliable, sensitive, and rapid detection of strains of *A. citrulli* causing BFB in melon. These markers may also be useful in early disease detection in the field samples, in seed health tests, and for international quarantine purposes.

## 1. Introduction

*Acidovorax citrulli*, formerly *Acidovorax avenae* subspecies *citrulli* (*A. citrulli*), is an aerobic, mesophilic, gram-negative, and seed-borne bacterium belonging to the beta subdivision of the Proteobacteria [1]. This bacterial pathogen causes bacterial fruit blotch (BFB) disease [1,2] and represents a serious threat to cucurbit crops, mainly melon (*Cucumis melo* L., Cucurbitaceae) and watermelon (*Citrullus lanatus* L., Cucurbitaceae), worldwide, including South Korea [3,4,5,6]. The disease was first reported in Georgia, United States (US) in the mid-1960s. However, it gained importance in the late 1980s and early 1990s [7] when severe outbreaks of the disease caused complete yield losses in commercial watermelon fruit production fields in several states of the US [8,9,10]. In fact, extensive infections in the mid-1990s incited a period of “high stakes” lawsuits against seed companies by growers, causing a near shutdown of watermelon production in the US in 1995 [11].

Since then, BFB disease has been reported in 22 countries [4] and is considered the most serious threat to watermelon, melon, and other cucurbit crops such as citron melon, prickly paddy melon, pumpkin, cucumber, squash, and several types of gourds in many regions of the world [3,12,13,14,15,16,17]. In melon, BFB is estimated to cause 80% to 100% loss in production under favorable environmental conditions, especially during rainy seasons and highly fluctuating temperature regimes [18,19,20]. In South Korea, BFB in melon was first reported in 1990, and several outbreaks have been observed in recent years [21]. 

The global spread of the disease can be attributed to the inadvertent distribution of contaminated commercial seeds, infected transplants, or alternative hosts and to the changes in population structure of the causal agent [4,15,22,23]. To prevent distribution of the disease, many countries, including the United States, China, and Europe, have already imposed import restrictions on contaminated seeds and plant materials [24]. In addition, use of pathogen-free seeds and transplants is recommended, as chemical control of infected plants is only partly or moderately successful [23]. Early detection of the pathogen from asymptomatic fields is likely to be helpful in employing timely control measures against the disease. Thus, the development of efficient, reliable, and sensitive diagnostic tools for detecting *A. citrulli* strains is necessary [24]. *A. citrulli* strains were previously detected by differential or semi-selective agar media [25,26], seedling grow-out, sweat box or dome assays [27], carbon source utilization profiles, fatty acid methyl esters and serological assays [28,29,30,31,32,33,34,35], and matrix-assisted laser desorption/ionization time-of-flight mass spectrometry (MALDI-TOF MS) and Fourier transform infrared (FTIR) spectra [36]. A self-paired colloidal gold immune chromatographic test strip (Sa-GICS) [37], Raman hyperspectral imaging [38], surface plasmon resonance (SPR) imaging [34], lateral flow immune chromatographic strip (ICS) [39], cross-priming amplification (CPA)-based isothermal DNA amplification [40], and visual loop-mediated isothermal amplification (LAMP) [41] also used for *A. citrulli* strains detection. Although these techniques have unique advantages, they can be expensive, laborious, and lengthy; may require extensive prior knowledge of the morphological and biochemical properties of the strains. Sometimes skilled manpower, and specialized equipment’s are needed during detection of closely related strains [42].

Polymerase chain reaction (PCR)-based assays, on the other hand, are advantageous as they offer reliable, quick, and sensitive detection of pathogenic bacterial agents. Several PCR-based assays have been successfully used for detection of *A. citrulli* strains, such as classical PCR; dye-based quantitative PCR, ethidium monoazide (EMA)-PCR [43], propidium monoazide (PMA)-PCR [24]), SYBR green-based real-time PCR [42,44], andenterobacterial repetitive intergenic consensus polymerase chain reaction (ERIC-PCR) [44,45].Contrastingly, PCR assays combined with immunological techniques [46,47], immune magnetic separation (IMS) [48], and magnetic capture hybridization [49] additionally used for the detection of *A. citrulli* strains. These PCR assays were developed to target specific genes, such as 16S rDNA, internal transcribed spacer (ITS) regions, YD-repeat proteins, and *hrp* genes, which often lack substantial polymorphism between closely related bacterial species and thus may not be suitable for the detection of host-crop-wise virulent *A. citrulli* strains [50,51,52]. Recently, whole genome sequence of *A. citrulli* strains were reported in melon [50,53] and watermelon [54,55,56]. Most of the assays were developed to detect *A. citrulli* strains at the species level [42,45,57,58,59,60] or to distinguish model group II strain AAC00-1 (more virulent to watermelon hosts) from model group I strain M6 (more virulent to all non-watermelon cucurbit hosts) [50,61].

Globally, melon production is suffering because of BFB disease. So far, no molecular marker is available for detecting *A. citrulli* strains that cause BFB in melon. In this study, we aimed to develop PCR-based novel molecular markers for detecting *A. citrulli* strains causing BFB in melon. For designing such primers, we used a whole-genome alignment-based approach to identify genomic fragments that are only present in *A. citrulli* strains that are particularly virulent towards melon. 

## 2. Results

### 2.1. Pathogenicity of Acidovorax citrulli Strains NIHHS15-280 and KACC18782

The comparative pathogenicity of two Korean *A. citrulli* strains, NIHHS15-280 and KACC18782 (isolated from melon; Table 1), was tested on different cucurbit crops such as melon, watermelon, cucumber, and pumpkin using a cross-inoculation assay (Figure 1).

Typical BFB symptoms were observed in NIHHS15-280- and KACC18782-inoculated leaves of the susceptible melon genotype PI614596 at 12 days after inoculation (DAI), whereas no/mildly symptoms were visible on watermelon, cucumber, or pumpkin leaves (Figure 1a,b). When inoculated with their respective known virulent strains, NIHHS16-088, KACC17002, and KACC17913, typical BFB symptoms developed on watermelon, cucumber, and pumpkin genotypes, respectively (Figure 1c). Cross-inoculation of these strains on melon (PI614596) did not/mildly produce BFB symptoms (Figure 1d).

*A. citrulli* strain NIHHS15-280 also caused BFB symptoms on melon fruits at 7 DAI (Figure 1e). Overall, the appearance of typical BFB symptoms on melon leaves only, with no/somewhat visible disease symptoms on watermelon, pumpkin, or cucumber leaves, indicated that strains NIHHS15-280 and KACC18782 were highly virulent towards the melon.

### 2.2. Whole-Genome Alignment Allows Design of Melon Host-Specific and A. citrulli Strain Specific Primers

The genome sizes of the three “*Ac*WM” strains (5.07516–5.35277 Mb) were slightly higher than that of the three “*Ac*M” strains (4.82187–4.90344 Mb). However, the GC contents were similar among the three “*Ac*WM” strain and the two “*Ac*M” strain (68.53–68.87%) excluding DSM17060. In case of all *A. citrulli* number of contigs varied from 1 to 139, number of 16S rRNA gene is 4 to 5, Symmetric Identity (%) also varied from 90.48 to 99.98 (apart from KACC17005) and Gapped Identity (%) almost similar (not including KACC17005) (Table 2).

The whole-genome alignment of these bacterial strains revealed that most nucleotide sequences are conserved; along with multiple rearrangements were observed among these pathogenic strains (Figure 2a–d).

The whole-genome alignment facilitated the identification of genomic fragments that are present only in “*Ac*M”. Three sets of primers, *AcM13*, *AcM380*, and *AcM797*, were designed from the three “*Ac*M” strains specific on local colonial blocks (LCB) of LCB13, LCB380, and LCB797, respectively. These primers were expected to produce 724, 853- and 938-bp amplicons, respectively, from “*Ac*M” strains (Figure 2a,b and Table 3). In addition, two sets of primers, *Ac33* and *Ac1410*, were designed from on LCBs of LCB33 and LCB1410, which were common to *A. citrulli* strains specific. These primers were expected to amplify 874 and 699-bp genomic fragments from all six *A. citrulli* strains, irrespective of their association with melon or watermelon hosts (Figure 2c,d and Table 3).

### 2.3. PCR-Based Assays Show Specificity of Primers

Three primer sets *AcM13*, *AcM380*, and *AcM797*, designed for detecting “*Ac*M” strains, produced the anticipated PCR amplicons of 724, 853, and 938-bp, respectively, for both the NIHHS15-280 and KACC18782 strains that were highly virulent towards melon (Figure 3a). *A. citrulli* strains that were associated with additional cucurbit (such as watermelon, cucumber, and pumpkin) and other bacterial strains such as *Acidovorax avenae* subsp. *avenae* (KACC16207), *Pseudomonas syringae* pathovar. *maculicola* (ICMP13051), and the fungal species *Didymella bryoniae* (NIHHS1326) were not amplified (Figure 3a) by these “*Ac*M” strain-specific primers. Two *A. citrulli* strain specific-primers *Ac33* and *Ac1410* designed for detecting *A. citrulli* strains, irrespective of their host, detected all target *A. citrulli* strains but not the other bacterial and fungal strains (Figure 3b).

### 2.4. Bio-PCR Shows Sensitivity of the Primers

The sensitivity of the developed primers for detection of “*Ac*M” strains from artificially infected samples was tested using Bio-PCR. Primer sets *AcM13*, *AcM380*, and *AcM797* successfully produced amplicons of the predictable sizes from leaf (Figure 4a) and fruit/seed (Figure 4b) washing suspensions of melon infected with NIHHS15-280 strain. No amplification was observed for leaf washings of watermelon, cucumber or pumpkin (Figure 4a), in agreement with the lack of observed disease symptoms (Figure 1a). However, when inoculated with their respective virulent *A. citrulli* strains, conspicuous BFB symptoms were visible on watermelon, cucumber, and pumpkin leaves, and the bacteria were successfully detected by *A. citrulli* specific-primer sets *Ac33* and *Ac1410* in the Bio-PCR assay (Figure 4c).

### 2.5. Detection Limits of the PCR Assays

The minimum DNA concentration of *A. citrulli* strains required for PCR detection using our primers was determined with a 10-fold dilution series, ranging from 65 ng/µL to 6.5 × 10^−3^ ng/µL, of gDNA of *A. citrulli* NIHHS15-280.The minimum detection limits of all strain-specific primer sets i.e. “*Ac*M” (*AcM13*, *AcM380*, and *AcM797*) and “*Ac*WM” (*Ac33* and *Ac1410*) are 6.5 × 10^−3^ ng/µL gDNA of *A. citrulli* strain NIHHS15-280 (Figure 5a,b). These results were achieved using 25 cycles of PCR amplification of 1 μL of each series diluted bacterial gDNA following the condition mention in Section 4.5.

## 3. Discussion

The development of efficient diagnostic tools for bacterial fruit blotch-causing *A. citrulli* strains has been the subject of intense investigation over the past few decades. Substantial population diversity exists among strains of *A. citrulli*, and strains are known to show differential virulence towards various cucurbit hosts [5,23,33,45,50,61,64]. Based on this differential virulence, DNA finger-printing and biochemical properties have characterized two major groups of *A. citrulli*, with group II strains being predominantly isolated from watermelon hosts, while group I strains are associated with non-watermelon hosts [35]. Each strain shows aggressive virulence to its corresponding hosts [45,65]. This defined group II strains are more virulent towards watermelon. The differential virulence of group I strains to various non-watermelon cucurbit hosts, such as melon, cucumber, pumpkin, and gourds, is yet to be clearly characterized. This is particularly important, as reports of host range expansion, global spread, and increasing virulence of the pathogen suggest that *A. citrulli* strains of group I could also be differentially virulent to various non-watermelon cucurbit crops [4,23,66]. 

Among non-watermelon cucurbit crops, melon production is known to be greatly affected by BFB. After watermelon, melon is an important fruit crop, with 26 million tons of melons produced global in 2009 (http://faostat.fao.org). There are reports of many *A. citrulli* strains isolated from melon hosts [5,52,61,64,67,68,69]. However, it is not known if these strains are specifically virulent towards melon or more virulent towards them. Our cross-inoculation assay with two such strains, NIHHS15-280 and KACC18782, on melon, watermelon, cucumber, and pumpkin revealed that these two strains are highly virulent towards melon (Figure 1). We then developed PCR-based markers for detecting such melon host-associated *A. citrulli* strains. 

Various techniques have been used for detecting *A. citrulli* strains, as discussed in Section 1. In general, PCR-based techniques are particularly advantageous as they detect strains based on inherent gDNA and are thus sensitive and specific towards the target strains only [42,50,51]. Most of the PCR-based *A. citrulli* strain-specific markers were developed by targeting specific genes, such as 16S rDNA [48], 16S-23S ITS regions of rDNA [26,28,70], YD-repeat protein [44], BOX fragments [58,67], and *hrp* gene sequences [71,72]. These genes are highly conserved among the members of a particular bacterial species and thus are successful in distinguishing *A. citrulli* strains from other bacterial species [42,45,57,58,59,60]. However, when it comes to distinguishing closely related members of the same species, for example in our case, detecting *A. citrulli* strains that are more virulent towards melon hosts, primers need to be designed from genomic regions that exist only in the target strains. 

The availability of whole-genome sequences of an increasing number of bacterial strains (mainly due to the advances and affordability in genome sequencing technologies) and the development of comprehensive suites of sequence analytical tools offers new opportunities in identifying strain-specific genomic regions by comparing the whole genomes of closely related bacterial strains. Such approaches have been successfully used for developing race- and pathovar-specific molecular markers for *Xanthomonas campestris* pv. *campestris* [73,74,75], *Xanthomonas euvesicatoria* [76], *Cordyceps militaris* [77], and *Plasmodiophora brassicae* [78]. In the case of *A. citrulli*, Eckshtain-Levi et al. [50] successfully used this comparative genomic approach to identify eight genomic fragments that are only present in model group II *A. citrulli* strain AAC00-1 and absent in model group I *A. citrulli* strain M6. 

We compared whole-genome sequences of three *A. citrulli* strains M6, pslb65, and DSM 17060, that were isolated from melon with three *A. citrulli* strains AAC00-1, KACC17005, and Tw6, that were isolated from watermelon (Table 1 and Figure 2) to identify differential genomic regions. Three genomes of each bacterial group were used to ensure the identification of robust strain-specific differential genomic regions. Three sets of primers, *AcM13*, *AcM380*, and *AcM797*, were designed from the unique genomic fragments hosting genes encoding glycosyl hydrolase, ATP-binding protein, and twin-arginine translocation pathway protein, respectively, that were only present in “*Ac*M” strains and absent in “*Ac*WM” strains (Table 3). The ATP-binding proteins are essential for cell viability, virulence, and pathogenicity [79,80], and twin-arginine translocation pathway proteins are known to be very important for bacterial virulence [81,82,83]. The glycosyl hydrolases are known to play roles in bacterial pathogenesis in mammals, especially by producing enzymes that can modify the carbohydrates of host defense mechanisms to favor bacterial survival and persistence [84,85,86]. However, the exact role of glycosyl hydrolases in phytopathogenic bacteria is yet to be known [85].

Three primer sets successfully detected “*Ac*M” strains during the validation test, while the strains associated with other cucurbit hosts such as watermelon, cucumber, and pumpkin and strains of other bacterial species were not. These primers also detected the *A. citrulli* strains (melon host) from artificially infected leaf, fruit and seed washing suspensions in a Bio-PCR assay, which does not require the isolation of bacterial DNA. The ability to detect bacteria from a very low concentration of template DNA further indicated the high sensitivity of the developed markers. However, the efficacy of the developed markers should be further validated using a wider range of *A. citrulli* strains associated with different cucurbit hosts.

## 4. Materials and Methods

### 4.1. Retrieval and Alignment of Whole Genomes of A. citrulli Strains

Whole-genome sequences of three *A. citrulli* strains M6, Pslb65, and DSM 17060 that were isolated from melon hosts and three *A. citrulli* strains AAC00-1, KACC17005, and tw6 that were isolated from watermelon hosts (referred to as “*Ac*M” and “*Ac*WM” strains respectively, throughout the manuscript) were retrieved from the National Center for Biotechnology Information database (https://www.ncbi.nlm.nih.gov/). The details of these strains are given in Table 1. The whole-genome sequences were aligned using the multiple sequence alignment tools Mauve, version v.2.4.0 (http://darlinglab.org/mauve/download.html), and Geneious (14 days free trial version; https://www.geneious.com/free-trial/) (Figure 2) with default parameters to identify common and unique genomic fragments between *A. citrulli* strains associated with melon and watermelon hosts. 

### 4.2. Design of Primers for Detection of A. citrulli Strains

Three pairs of primers (*AcM13*, *AcM380*, and *AcM797*) for detecting *A. citrulli* strains associated with melon hosts were designed from three segments (269,247–269,971 bp, 1,911,322–1,912,175 bp, and 271,733–272,671 bp) in the M6 strain (marked as red rectangles in Figure 2a,b), respectively, that are only present in “*Ac*M” strains (Figure 2a,b). In addition, two sets of *A. citrulli* specific-primers (*Ac33* and *Ac1410*) were designed from the segments 3,879,509–3,880,383 bp and 3,120,584–3,121,283 bp in the AAC00-1 strain (green rectangles in Figure 2c,d), respectively, that are present in both “*Ac*M” and “*Ac*WM” strains (Figure 2c,d). Primers were designed using Primer3 software (http://www.bioinformatics.nl/cgi-bin/primer3plus/primer3plus.cgi), and in silico specificity testing was done using the “in silico simulation of molecular biology experiments” tool accessible from http://insilico.ehu.eus/. The details of the primers along with amplicon size, associated genes, and PCR conditions are given in Table 3.

### 4.3. Bacterial Strains, Growing Conditions, and Inoculum Preparation

Nineteen bacterial strains belonging to three different genera, *Acidovorax*, *Pseudomonas*, and *Didymella*, were collected from various sources for this study. The details of the strains along with their hosts and collection source are given in Table 1. All bacteria were cultured on King’s B (KB) media supplemented with 100 µg·mL^−1^ ampicillin for 36 to 48 h at 28 °C until formation of bacterial colonies. All bacterial suspensions were prepared by inundating the culture plates with 5 mL of sterile, double-distilled water (DDW) and gently scraping off the bacterial colonies using a sterile L-shaped rubber spreader. For inoculation, bacterial suspensions were diluted to a final concentration of ~3 × 10^6^ colony-forming units (cfu) mL^−1^.

### 4.4. Extraction of Bacterial Genomic DNA

Genomic DNA (gDNA) from all bacterial strains was extracted using a DNeasy Plant Mini Kit (Qiagen, Hilden, Germany) following the manufacturer’s instructions. The amount and quality of the extracted DNA was then measured using a Nanodrop ND-1000 spectrophotometer (Nano Drop, Wilmington, DE, USA).

### 4.5. Specificity of Primers in PCR Assays

The specificity of the three “*Ac*M” strains-specific primers, *AcM13*, *AcM380*, and *AcM797*, and two *A. citrulli* strain-specific primers, *Ac33* and *Ac1410*, was assessed in a PCR assay using gDNA from the target melon host *A. citrulli* strains NIHHS15-280 and KACC18782, along with other cucurbit host-specific *A. citrulli* strains and other bacterial strains. PCR was performed in a 20-μL reaction mixture containing 9 µL of Emerald PCR master mix (Takara, Shiga, Japan), 1 µL each of forward and reverse primer (10 picomole), 8 µL of ultrapure water, and 1 µL of bacterial gDNA (~50 ng/µL). PCR was performed in a thermocycler (Takara) using the following conditions: initial denaturation at 95 °C for 5 min followed by 25 cycles of denaturation at 95 °C for 30 s, annealing at 65 °C for 45 s (for “*Ac*M” strain-specific primers *AcM13*, *AcM380*, and *AcM797*) or 63 °C for 30 s (for *A. citrulli* strain-specific primers *Ac33* and *Ac1410*), then 72 °C for 40 s, and a final elongation at 72 °C for 5 min (Table 3). Electrophoresis was performed using a 1.2% agarose gel stained with blue mango (BioD, Gwangmyeong, Korea) in TAE buffer at 100 V for 40 min and visualized on an ENDURO™ GDS gel documentation system under UV radiation (350 nm). *A. citrulli* strain-specific primers *Ac33* and *Ac1410* were used as positive controls to identify any potential false-positive results. DDW was used as a negative control. HIQ 100 bp plus DNA Ladder Mix (BIONEER) was used as a size marker.

### 4.6. Pathogenicity Test of Bacterial Strains Using Cross Inoculation Assay

Susceptible cultivars of melon (PI614596) [87], watermelon (Charleston Gray) [88], cucumber (PI200815), and pumpkin (Kabocha) were grown in 32-celled trays containing artificial soil mix in a controlled plant growth chamber at 25 ± 2 °C, with a 16-h day length and 440 µmoles/m^2^/s light intensity at the bench level in a glasshouse. Five-week-old plants were inoculated with a concentration of ~3 × 10^6^ (cfu) mL^−1^ [22] with *A. citrulli* strains NIHHS15-280 and KACC18782 (collected from melon hosts, as per the records of Korean Agriculture Culture Collection (KACC) and NIHHS), as shown in Figure 1a,b. These genotypes were also inoculated with corresponding *A. citrulli* strains NIHHS16-088, KACC17002, and KACC17913 that are virulent to watermelon, cucumber, and pumpkin, respectively (Figure 1c). Furthermore, melon (PI614596) lines again inoculated with *A. citrulli* strains NIHHS16-088, KACC17002 and KACC17913 that are virulent to watermelon, cucumber & pumpkin, respectively (Figure 1d). In addition, mature detached fruits of melon were inoculated by injecting 1 ml of bacterial suspension at 1 × 10^8^ cfu/mL [23] through the rind with a 5-ml syringe, at a depth of no more than 1 cm, in two different sites per fruit with *A. citrulli* strain NIHHS15-280. Inoculated fruits were placed in plastic boxes (one fruit/box) and incubated at 25 to 28 °C. Disease symptoms on both leaves and fruits were recorded at 12 and 7 DAI, respectively. The leaves, fruit and seed washing suspensions were used as template DNA for detection of *A. citrulli* strains causing BFB on melon by the developed markers in a Bio-PCR-based assay, as described in Section 4.7. The experimental design was completely randomized with three replicates (plants) per genotype, with three leaves and fruits assessed per replicate.

### 4.7. Bio-PCR Assay 

For the Bio-PCR assay, infected leaf, fruit and seeds washings (of samples from the cross-inoculation assay as shown in Figure 1) were used as template DNA. Small, dark-brown, water-soaked, visible lesions of about 2 cm were cut into small pieces using a sterile scalpel and immersed in 500 µL of sterile distilled water in a 1.5-mL tube for 40 min (with occasional shaking). Thereafter, 10 µL of washing suspension (containing bacterial cells) was collected for the Bio-PCR reaction. The Bio-PCR assay was performed using the “*Ac*M” strain-specific primers *AcM13*, *AcM380*, and *AcM797* and *A. citrulli* strain-specific primers *Ac33* and *Ac1410* using the same PCR conditions as described in Section 4.5, except that the number of amplification cycles was 30. Increasing the number of PCR cycles may detect target strains at an even lower concentration of template DNA.

### 4.8. Detection Limit of Bacterial DNA in the PCR Assay

The minimum gDNA concentration of *A. citrulli* strains detected by the developed primers in the PCR assay was determined using a 10-fold dilution series (from 65 to 6.5 × 10^−4^ ng/µL) of the gDNA of *A. citrulli* strain NIHHS15-280. One microliter of each dilution was used directly as the template DNA with PCR conditions as described in Section 4.5.

## 5. Conclusions

This study exploited the variation within the whole-genome sequences of *A. citrulli* strains to develop PCR-based markers for detecting *A. citrulli* strains that are associated with melon hosts. This is the first report of PCR-based markers that can be used for direct, sensitive, and rapid detection of the *A. citrulli* strains that cause BFB disease, particularly in melon. These markers have the potential to detect melon host *A. citrulli* strains from asymptomatic fields prior to the appearance of symptoms, allowing farmers to enact timely control measures. The markers could also allow seed health testing prior to planting and ensure transboundary movement of disease-free seeds for quarantine purposes. 

## Figures and Tables

**Figure 1 ijms-20-02715-f001:**
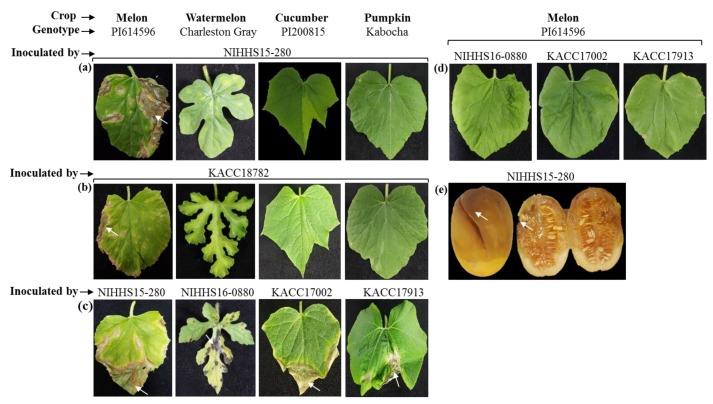
Melon, watermelon, cucumber, and pumpkin plants were inoculated with *A. citrulli* strains (**a**) NIHHS15-280, (**b**) KACC18782, (**c**) corresponding host, (**d**) melon with NIHHS16-088, KACC17002, and KACC17913 strains and (**e**) melon fruits with NIHHS15-280 strain. *A. citrulli* strains NIHHS15-280 and KACC18782 showing exclusively virulence towards melon leaves and fruits at 12 and 7 DAI respectively. Typical BFB symptoms are indicated by white arrows. All leaves and fruits were detached just before taking photographs.

**Figure 2 ijms-20-02715-f002:**
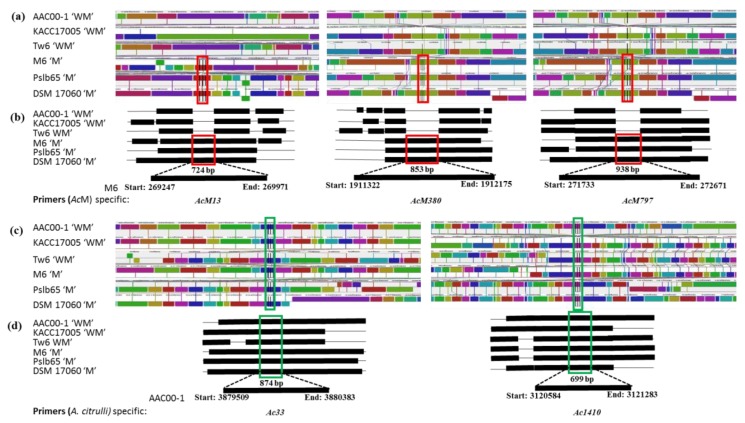
Segments of whole-genome alignment of six *A. citrulli* strains. Primers *AcM13*, *AcM380*, and *AcM797* were designed from the M6 strain where three segments (red rectangles) that are present only in “*Ac*M” strains (**a**,**b**). *A. citrulli* specific-primers *Ac33* and *Ac1410* were designed from the AAC001 strain where two segments (green rectangles) that are present in both “*Ac*WM” and “*Ac*M” strains (**c**,**d**). Details of the primers are shown in Table 3. In the Mauve alignment (Mauve tool, version 2.4.0), each genome is laid out horizontally with homologous segments (LCBs) outlined as colored rectangles. Lines collate aligned segments among genomes. Average sequence similarities within an LCBs, measured in sliding windows, are proportional to the heights of interior colored bars. Large Sections of white within blocks and gaps between blocks indicate lineage-specific sequences. Right side in strains name WM and M indicate *A. citrulli* strains isolated from watermelon and melon hosts, respectively.

**Figure 3 ijms-20-02715-f003:**
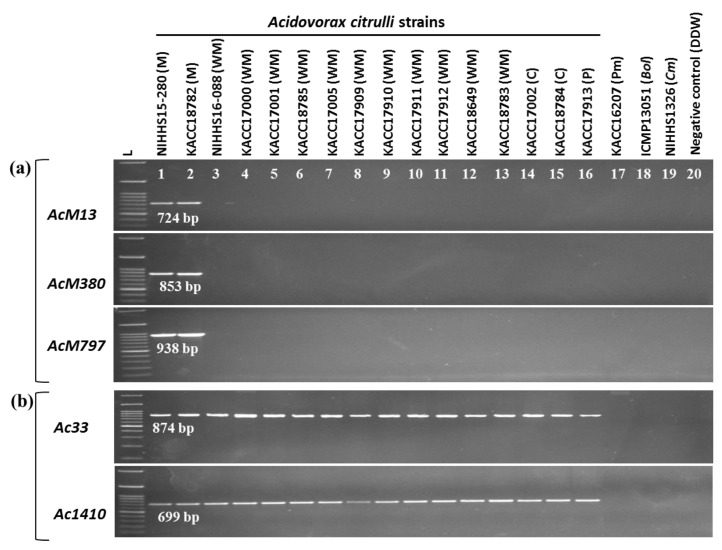
Agarose gel electrophoresis of polymerase chain reaction (PCR) products from gDNA of *A. citrulli* strains, other bacterial strains, and a fungal isolate using the primers designed based on whole-genome alignment. PCR using “*Ac*M” strain-specific primers *AcM13*, *AcM380*, and *AcM797* (**a**) and *A. citrulli* strain-specific primers *Ac33* and *Ac1140* (**b**). gDNA (~50 ng/µL) of the bacterial strains was used as template. *A. citrulli* strains-specific primers *Ac33* and *Ac1410* were used as controls to check the PCR amplification of the bacterial DNA. Strains followed by the letters M, WM, P, C, Pm, *Bol*, and *Cm* indicate pathogenic strains reported to be collected from melon, watermelon, pumpkin, cucumber, proso-millet, *Brassica oleracea* var. *capitata*, and *Cucumis melo* hosts, respectively. Non-*A. citrulli* strains such as KACC16207 (*Acidovorax avenae* subsp. *Avenae*), ICMP13051 (*Pseudomonas syringae* pv. *maculicola*) and a fungal isolate NIHHS1326 (*Didymella bryoniae*), as well as DDW, were used as negative controls. Lane L indicates a HIQ 100 bp plus DNA Ladder Mix (BIONEER, Korea) used as a size marker.

**Figure 4 ijms-20-02715-f004:**
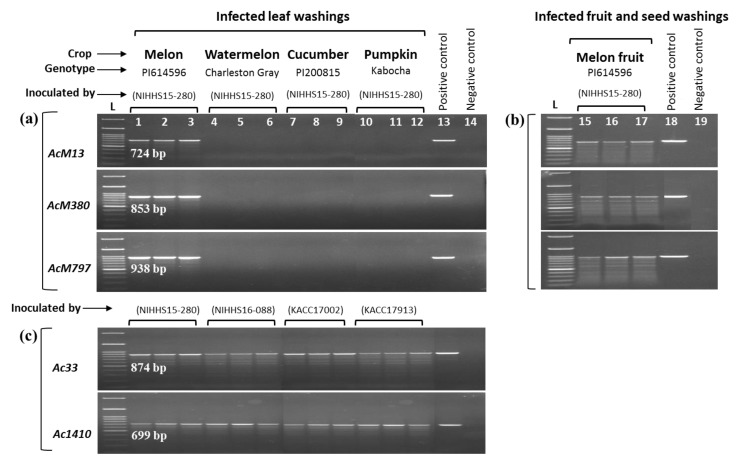
Bio-PCR detection of *A. citrulli* from the artificially infected leaf, fruit and seed washings using the primers designed based on whole-genome alignment. (**a**) Artificially infected (inoculated with NIHHS15-280) leaf and (**b**) fruit/seed washings were used as PCR templates for the “*Ac*M” strain-specific primer sets AcM13, AcM380 and AcM797. *A. citrulli*-strain specific primer sets *Ac33* and *Ac1410* detected all *A. citrulli* strains from the leaf washings of melon, watermelon, cucumber, and pumpkin infected by the virulent strains NIHHS15-280, NIHHS16-088, KACC17002, and KACC17913, respectively (**c**). Three independent samples for each crop–strain combination were assessed. gDNA of *A. citrulli* strain NIHHS15-280 (lanes 13, 18) and double distilled water (lanes 14, 19) were used as positive and negative controls, respectively. HIQ 100 bp plus DNA Ladder Mix (BIONEER) was used as a size marker (lane L). Details of the inoculation procedure are described in Section 4.5, and collection of leaf washings and Bio-PCR procedures are described in Section 4.7.

**Figure 5 ijms-20-02715-f005:**
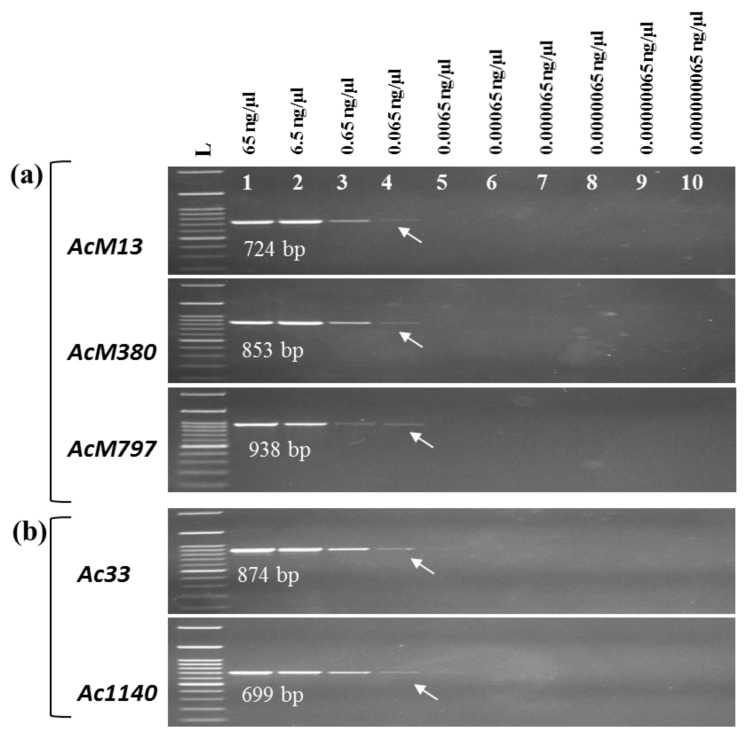
DNA detection limits of the PCR assays. Detection limit of gDNA of *A. citrulli* strain using “*Ac*M” strain-specific primer sets (**a**) *AcM13*, *AcM380*, and *AcM797* and (**b**) *A. citrulli* strain-specific primer sets *Ac33* and *Ac1410* in PCR assays. Lanes 1 to 10, 10-fold dilution series of gDNA of *A. citrulli* strain NIHHS15-280 starting from 65 ng/µL; Lane L contains HIQ 100 bp plus DNA Ladder Mix (BIONEER, Republic of Korea) as a size marker.

**Table 1 ijms-20-02715-t001:** Plant pathogenic bacterial strains of *Acidovorax citrulli* and other bacterial genera used in this study.

Bacterial Strains	Scientific Name	Host Plant	SC
NIHHS15-280, KACC18782	*Acidovorax citrulli*	Melon	South Korea
NIHHS16-088, KACC17000, KACC17001, KACC18785, KACC17005, KACC17909, KACC17910, KACC17911, KACC17912, KACC18649, KACC18783	*Acidovorax citrulli*	Watermelon	
KACC17913	*Acidovorax citrulli*	Pumpkin seed	
KACC17002, KACC18784	*Acidovorax citrulli*	Cucumber	
KACC16207	*Acidovorax avenae* subsp. *avenae*	Proso- millet	
ICMP13051	*Pseudomonas syringae* pv. *maculicola*	Cabbage	New Zealand
NIHHS1326	*Didymella bryoniae*	*Cucumis melo*	South Korea

ICMP: International Collection of Microorganisms from Plants (ICMP), Landcare Centre, Auckland, New Zealand; KACC: Korean Agricultural Culture Collection, Korea; NIHHS: National Institute of Horticultural and Herbal Science, Korea; SC: Source Country.

**Table 2 ijms-20-02715-t002:** Genomic information of *A. citrulli* strains used for whole-genome alignment.

SI	Strain	HP	GS (Mb)	GC%	Contigs	16S rRNA	BioSample	SI (%)	GI (%)	References
1	AAC00-1	Watermelon	5.35277	68.53	1	4	SAMN02598334	99.80	99.98	[54]
2	KACC17005	Watermelon	5.34992	68.54	1	4	SAMN07718226			[55]
3	Tw6	Watermelon	5.07516	68.70	24	5	SAMN03268415	90.48	99.52	[62]
4	M6	Melon	4.82187	68.87	139	5	SAMN04157986	91.36	99.63	[50]
5	pslb65	Melon	4.90344	68.80	24	5	SAMN03333326	92.04	99.54	[53]
6	DSM 17060	Melon	4.84827		71	5	SAMN04489709	92.04	99.51	[63]

N.B. The strains isolated from, and/or more virulent to, watermelon and melon hosts, are referred to as “*Ac*WM” and “*Ac*M” respectively, throughout the manuscript; HP, GS, SI, and GI denotes Host Plant, Genome Size, Symmetric Identity, and Gapped Identity, respectively.

**Table 3 ijms-20-02715-t003:** Primer sets designed for detecting *A. citrulli* strains causing BFB disease via polymerase chain reaction (PCR) assays.

Primer Name	Primer Sequence	LCB	Genomic Location	Gene Description	Amplicon Size (bp)	Annealing Condition
*AcM13*	F: TCGCGGGCCGTGATGTTCCG R: TGGACTTCGGGTGGGCCTTCA	13	(+) 269247 to 269971	Glycosyl hydrolase	724	65 °C for 45 s
*AcM380*	F: GCATCCGGTGTGCTGCTGGA R: GAGATGTCAGAGTCGCACGGT	380	(+) 1911322 to 1912175	ATP-binding protein	853	
*AcM797*	F: AAGGCGGACATGGGTTGGCT R: CTGCGCCTGCGCCCACACCA	797	(+) 271733 to 272671	Twin-arginine translocation pathway signal	938	
*Ac33*	F: TCGATAAGGCCACCAAGTTC R: GACTGGGGTAACGTGGGGCT	33	(−) 3879509 to 3880383	Hypothetical protein phosphatase	874	63 °C for 30 s
*Ac1410*	F: TAGCGCAGCCCCACCCAGTG R: CAAGGGCGACAAGATGATGT	1410	(+) 3120584 to 3121283	Sugar-binding protein	699	

*AcM13*, *AcM380*, and *AcM797* were designed from the genome sequence of *A. citrulli* strain M6 that is only present in “(*Ac*M)” and *Ac33* and *Ac1410* were designed from the *A. citrulli* strain AAC001, which is present in all *A. citrulli* strains; LCB, local colonial block; +/−, positive/negative strand; F and R, forward and reverse primers, respectively.

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
