# Peer review of "Development of Molecular Markers for Detection of Acidovorax citrulli Strains Causing Bacterial Fruit Blotch Disease in Melon"

_ijms, 2019, doi:10.3390/ijms20112715_

Round 1
Reviewer 1 Report
Comments and Suggestions for Authors
This manuscript titled ‘Whole genome alignment based development of molecular markers for detection of Acidovorax citrulii strains causing bacterial fruit blotch disease in melon’ advance our understanding on variation in whole-genome sequences of A. citrulli which helps develop PCR based markers.
Overall, this is an interesting study and it provides a reliable and rapid detection of A. citrulli bacterial strains that causes fruit blotch in melon.
I suggest the authors to change the title to Detection of Acidovorax citrulii strains causing bacterial fruit blotch disease in melon or other similar title, because this explains the main focus of the experiment. Including ‘whole genome alignment’ as first words is slightly misleading.
The Tables 1, 2 and 3 needs to be sized and spaced correctly. The information provided is relevant but the spacing is completely off.
Please double check the reference format and make sure it follows the journal guidelines. The scientific names of organisms are not italicized in numerous places in the references section. Please double check the references format. Additionally, the manuscript has several long sentences (Eg: lines 57-65; 65-69; 70-76, etc.). Please rephrase them. I suggest that authors have the manuscript checked by a native English-speaking colleague or use a professional English editing service.
Author Response
Response to Reviewer 1 comments
General comments to the manuscript:
Comments: This manuscript titled ‘Whole genome alignment based development of molecular markers for detection of Acidovorax citrulii strains causing bacterial fruit blotch disease in melon’ advance our understanding on variation in whole-genome sequences of A. citrulli which helps develop PCR based markers. Overall, this is an interesting study and it provides a reliable and rapid detection of A. citrulli bacterial strains that causes fruit blotch in melon. I suggest the authors to change the title to Detection of Acidovorax citrulii strains causing bacterial fruit blotch disease in melon or other similar title, because this explains the main focus of the experiment. Including ‘whole genome alignment’ as first words is slightly misleading
Responses: Thanks for your compliments. Title modified according to your valuable suggestions.
Comments: The Tables 1, 2 and 3 needs to be sized and spaced correctly. The information provided is relevant but the spacing is completely off.
Responses: Revised according to the Journal format. Thanks
Comments: Please double check the reference format and make sure it follows the journal guidelines. The scientific names of organisms are not italicized in numerous places in the references section. Please double check the references format. Additionally, the manuscript has several long sentences (Eg: lines 57-65; 65-69; 70-76, etc.). Please rephrase them. I suggest that authors have the manuscript checked by a native English-speaking colleague or use a professional English editing service.
Responses: Reference format checked and corrected accordingly. We divided the long sentences to short one. The manuscript already has been edited by Plant Editors, USA. Thanks for your observation.
Reviewer 2 Report
In this paper, Islam and co-workers develop a molecular marker for the detection of Acidovorax citrulli in melon. The article is scientifically sound, and it has a practical application to support melon producers in the early detection of a devastating bacterial pathogen. The materials and methods section requires some work, in particular, the formatting of the tables must be revised. The remaining sections fulfil the quality standard for publication. In the following lines you will find some suggestions to improve the already good quality of the manuscript:
Title:
If possible, remove “Whole-genome alignment-based” as it makes the title unnecessarily long.
citrulli should not be capitalised
Abstract:
Line 13: Include abbreviation BFB, after bacterial fruit blotch.
Line 19: Please state the name of the primer sets developed.
Line 22: Clarify that this was only done for one strain (NIHHS15-280) – No evidence was presented for the other strain.
Introduction:
Line 58 and 68: Avoid using “for example”, it is not required here.
Line 77: Change “internally” to “internal”.
Table 1: Please replace “host name” for “host plant”. Please modify formatting so table headings are within one line. Use abbreviations and explain them in the legend, e.g., source country could be SC.
Line 118: Please be consistent with the number of decimal places.
Table 2: Be consistent with the use of Mbp or Mb throughout the text, choose only one. Same as above, the headings should be aligned and within one line if possible (use abbreviations).
Line 129: In here, you mention that inversions, translocations and deletions were found. Please provide more detail in the text.
Figure 3 and 4: Please change –ve and +ve for either positive and negative, or + and -.
Figure 5. The minimum detectable limit is of 0.0065 for Ac33 and Ac1140 is extremely low, and cannot be seen clearly in the gel. The level should be increased to 0.065 or a level that is visible in the gel.
Line 212-13: When concentrations are mentioned, you talk about the cycles but there is no mention about the volume that went into the PCR, this should be mentioned here.
Discussion:
Line 279: If you had a PCR inhibitor, you would have no detection, so there would be no “false positive”. Please rephrase the sentence or delete it altogether.
Materials and methods:
In section 4.5. There seem to be missing steps in the protocol, especially in the denaturation step - it sounds like this was an initial denaturation, and you have forgotten to include an additional step before the annealing.
Acknowledgements:
Please give the full name for KACC and NIHHS.
Author Response
Response to Reviewer 2 comments
General comments to the manuscript:
Comments: In this paper, Islam and co-workers develop a molecular marker for the detection of Acidovorax citrulli in melon. The article is scientifically sound, and it has a practical application to support melon producers in the early detection of a devastating bacterial pathogen. The materials and methods section requires some work, in particular, the formatting of the tables must be revised. The remaining sections fulfil the quality standard for publication. In the following lines you will find some suggestions to improve the already good quality of the manuscript:
Responses: Thanks for your appreciated compliments. Tables are formatted according to the Journal requirement.
Comments: If possible, remove “Whole-genome alignment-based” as it makes the title unnecessarily long. citrulli should not be capitalised
Responses: Removed “Whole-genome alignment-based” from title and Citrulli corrected to citrulli. Thanks
Comments: Line 13: Include abbreviation BFB, after bacterial fruit blotch.
Responses: BFB Included as abbreviation. Thanks
Comments: Line 19: Please state the name of the primer sets developed.
Responses: Primer sets (AcM13, AcM380, and AcM797) are stated in parenthesis. Thanks
Comments: Line 22: Clarify that this was only done for one strain (NIHHS15-280) – No evidence was presented for the other strain.
Responses: We are sorry, because we use only as melon host representative strain (NIHHS15-280) for Bio PCR. So, we remove strain KACC18782 from Line 22. Thanks for your comments.
Comments: Line 58 and 68: Avoid using “for example”, it is not required here.
Responses: Deleted. Thanks
Comments: Line 77: Change “internally” to “internal”.
Responses: Changed. Thanks
Comments: Table 1: Please replace “host name” for “host plant”. Please modify formatting so table headings are within one line. Use abbreviations and explain them in the legend, e.g., source country could be SC.
Responses: Revised accordingly. Thanks
Comments: Line 118: Please be consistent with the number of decimal places.
Responses: Number of decimal places followed in consistent sequence. Thanks
Comments: Table 2: Be consistent with the use of Mbp or Mb throughout the text, choose only one. Same as above, the headings should be aligned and within one line if possible (use abbreviations).
Responses: Mbp changed to Mb for consistency. Headings aligned as one line and abbreviated long words. Thanks
Comments: Line 129: In here, you mention that inversions, translocations and deletions were found. Please provide more detail in the text.
Responses: As expected, across the length of the entire genomes of these pathogenic strains, there were some arrangements. Documenting all those arrangements would make the text and flow of the story incoherent. Instead of documenting all those, we focused on polymorphic regions (Fig. 2a-d) to develop markers. However, we revamped the statement in the manuscript (Line 129). Thanks
Comments: Figure 3 and4: Please change –ve and +ve for either positive and negative, or + and -.
Responses: We included positive and negative instead of –ve and +ve. Thanks
Comments: Figure 5. The minimum detectable limit is of 0.0065 for Ac33 and Ac1140 is extremely low, and cannot be seen clearly in the gel. The level should be increased to 0.065 or a level that is visible in the gel.
Responses: Corrected accordingly. Thanks
Comments: Line 212-13: When concentrations are mentioned, you talk about the cycles but there is no mention about the volume that went into the PCR, this should be mentioned here.
Responses: 1 μL of each series diluted bacterial gDNA volume are used (Included and highlighted in text). Thanks
Comments: Line 279: If you had a PCR inhibitor, you would have no detection, so there would be no “false positive”. Please rephrase the sentence or delete it altogether.
Responses: Deleted. Thanks
Comments: In section 4.5. There seem to be missing steps in the protocol, especially in the denaturation step - it sounds like this was an initial denaturation, and you have forgotten to include an additional step before the annealing.
Responses: Corrected according to the valuable suggestion by the reviewer. Thanks
Comments: Please give the full name for KACC and NIHHS.
Responses: In Acknowledgements section we included full name of KACC and NIHHS. Thanks